# WHEN DOES SELF-SUPERVISION IMPROVE FEW-SHOT LEARNING?

## ABSTRACT

We present a technique to improve the generalization of deep representations learned on small labeled datasets by introducing self-supervised tasks as auxiliary loss functions. Although recent research has shown benefits of self-supervised learning (SSL) on large unlabeled datasets, its utility on small datasets is unknown. We find that SSL reduces the relative error rate of few-shot meta-learners by 4%-27%, even when the datasets are small and only utilizing images within the datasets. The improvements are greater when the training set is smaller or the task is more challenging. Though the benefits of SSL may increase with larger training sets, we observe that SSL can have a negative impact on performance when there is a domain shift between distribution of images used for meta-learning and SSL. Based on this analysis we present a technique that automatically select images for SSL from a large, generic pool of unlabeled images for a given dataset using a domain classifier that provides further improvements. We present results using several meta-learners and self-supervised tasks across datasets with varying degrees of domain shifts and label sizes to characterize the effectiveness of SSL for few-shot learning.

## 1 INTRODUCTION

Current machine learning algorithms require enormous amounts of training data to learn new tasks. This is a problem for many practical problems across domains such as biology and medicine where labeled data is hard to come by. In contrast, we humans can quickly learn new concepts from limited training data. We are able to do this by relying on our past "visual experience". Recent work attempts to emulate this by training a feature representation to classify a training dataset of "base" classes with the hope that the resulting representation generalizes not just to unseen examples of the same classes but also to novel classes, which may have very few training examples (called few-shot learning). However, training for base class classification can force the network to only encode features that are useful for distinguishing between base classes. In the process, it might discard semantic information that is irrelevant for base classes but critical for novel classes. This might be especially true when the base dataset is small or when the class distinctions are challenging.

One way to recover this useful semantic information is to leverage representation learning techniques that do not use class labels, namely, *unsupervised* or *self-supervised learning*. The key idea is to learn about statistical regularities (*e.g.*, spatial relationship between patches, orientation of an images) that might be a cue to semantics. Despite recent advances, these techniques have only been applied to a few domains (*e.g.*, entry-level classes on internet imagery), and under the assumption that large amounts of unlabeled images are available. Their applicability to the general few-shot scenario described above is unclear. In particular, can these techniques help prevent overfitting to base classes and improve performance on novel classes in the few-shot setting? If so, does the benefit generalize across domains and to more challenging tasks? Moreover, can we use self-supervised training to boost performance in domains where even unlabeled images are hard to get?

This paper seeks to answer these questions. We show that with *no additional training data*, adding a self-supervised task as an auxiliary task (Figure 1) improves the performance of existing few-shot techniques on benchmarks across a multitude of domains (Figure 2). Intriguingly, we find that the benefits of self-supervision *increase* with the difficulty of the task, for example when training from a smaller base dataset, or with degraded inputs such as low resolution or greyscale images (Figure 3).

One might surmise that as with traditional SSL, additional unlabeled images might improve performance further. But what unlabeled images should we use for novel problem domains where unlabeled data is not freely available? To answer this, we conduct a series of experiments with additional unlabeled data from different domains. We find that adding more unlabeled images improves performance *only* when the images used for self-supervision are within the *same domain* as the base classes (Figure 4a); otherwise they can even *negatively* impact the performance of the few-shot learner (Figure 4b, 4c, 4d). Based on this analysis we present a simple approach that uses a domain classifier to pick similar-domain unlabeled images for self-supervision from a large, generic pool of images. The resulting method improves over the performance of a model trained with self-supervised learning from images within the domain (Figure 4c). Taken together, this results in a powerful, general and practical approach for improving few-shot learning from small datasets in novel domains. Finally, these benefits are also observed on standard classification tasks (Appendix A.3).

## 2  RELATED WORK

**Few-shot learning**  Few-shot learning aims to learn representations that generalize well to the novel classes where only few images are available. To this end several meta-learning approaches have been proposed that evaluate representations by sampling many few-shot tasks within the domain of a *base* dataset. These include optimization-based meta-learners, such as model-agnostic meta-learner (MAML) (Finn et al., 2017), gradient unrolling (Ravi & Larochelle, 2017), closed form solvers (Bertinetto et al., 2019), convex learners (Lee et al., 2019), *etc*. Others such as matching networks (Vinyals et al., 2016) and prototypical networks (Snell et al., 2017) rely on distance based classifiers, while Garcia & Bruna (2018) models higher-order relationships in data using a graph network. Another class of methods (Gidaris & Komodakis, 2018; Qi et al., 2018; Qiao et al., 2018) model the mapping between training data and classifier weights using a feed-forward network.

While the literature is rapidly growing, a recent study by Chen et al. (2019) has shown that the differences between meta-learners are diminished when deeper network architectures are used and they develop a strong baseline for few-shot learning based on ResNet18 architecture and prototypical networks Snell et al. (2017). We build our experiments on top of this work and show that auxiliary self-supervised tasks provide additional benefits with state-of-the-art deep networks across several few-shot benchmarks. We also find that prototypical networks are often the best performing, though the benefits of SSL generalize across meta-learners such as MAML.

**Self-supervised learning**  Human labels are expensive to collect and hard to scale up. To this end, there has been increasing research interest to investigate learning representations from unlabeled data. In particular, the image itself already contains structural information which can be utilized. Self-supervised learning approaches attempt to capture this.

There is a rich line of work on self-supervised learning. One class of methods removes part of the visual data, and tasks the network with predicting what has been removed from the rest in a discriminative manner (Larsson et al., 2016; Zhang et al., 2016; 2017). Doersch et al. (2015) proposed to predict the relative position of patches cropped from images. Wang & Gupta (2015) used the similarity of patches obtained from tracking as a self-supervised task and combined position and similarity predictions of patches (Wang et al., 2017). Other tasks include predicting rotation (Gidaris et al., 2018), noise (Bojanowski & Joulin, 2017), clusters (Caron et al., 2018; 2019; Wu et al., 2018), number of objects (Noroozi et al., 2017), missing patches (Pathak et al., 2016; Trinh et al., 2019), motion segmentation labels (Pathak et al., 2017), artifacts (Jenni & Favaro, 2018), next patch feature (Oord et al., 2018; Hénaff et al., 2019), or by maximizing mutual information (Hjelm et al., 2019; Bachman et al., 2019), and training an adversarial generative model (Donahue & Simonyan, 2019). Doersch & Zisserman (2017) jointly trained four self-supervised tasks and found it to be beneficial. Recently, Goyal et al. (2019) and Kolesnikov et al. (2019) compared various self-supervised learning tasks at scale and concluded that solving jigsaw puzzles and predicting image rotations are among the most effective, motivating the choice of self-supervised tasks in our experiments.

The focus of most prior works on self-supervised learning is to supplant traditional fully supervised representation learning with unsupervised learning on large unlabeled datasets for downstream tasks. Crucially in almost all prior works, self-supervised representations consistently lag behind fully-supervised representations trained on the same dataset with the same architecture (Goyal et al., 2019; Kolesnikov et al., 2019). In contrast, our work focuses on an important counterexample: self-

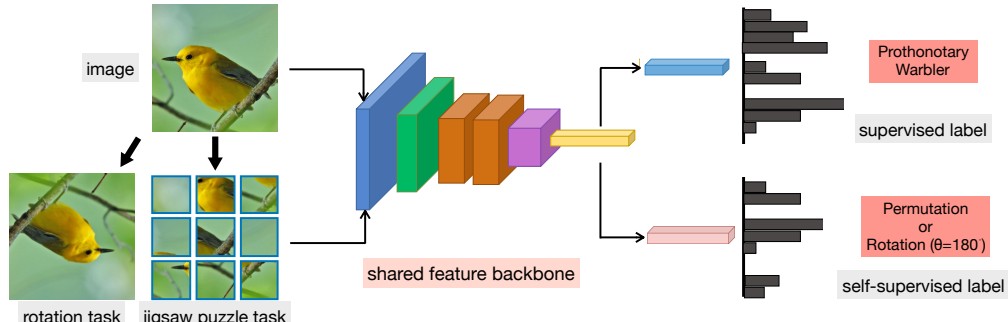

Figure 1: **Our framework combines supervised and self-supervised losses to learn representations on small datasets.** Self-supervised tasks, such as jigsaw puzzle or rotation prediction, can be derived from the images alone and act as a data-dependent regularizer for the shared feature backbone.

supervision can in fact augment standard supervised training for few-shot transfer learning in the low training data regime without relying on any external dataset. Our work also investigates the effect of domain shifts across datasets used for SSL and a given task.

**Multi-task learning** Our work is related to multi-task learning, a class of techniques that train on multiple task objectives together to improve each one. Previous works in the computer vision literature have shown moderate benefits by combining tasks such as edge, normal, and saliency estimation for images, or part segmentation and detection for humans (Kokkinos, 2017; Maninis et al., 2019; Ren & Lee, 2018). However, there is significant evidence that training on multiple tasks together often hurts performance on individual tasks (Kokkinos, 2017; Maninis et al., 2019). Only certain task combinations appear to be mutually beneficial, and sometimes specific architectures are needed. Our key contribution here is showing that self-supervised tasks and few-shot learning are indeed mutually beneficial in this sense. The fact that self-supervised tasks require no annotations makes our finding especially practically significant.

**Concurrent works** At the time of submission several concurrent works appear to draw similar conclusions to ours. We briefly describe these, deferring a detailed comparison to the future when code and experimental details become available. Asano et al. (2019) finds that self-supervised pre-training is useful even with a single image. However, very large images and extreme data augmentations are used. Carlucci et al. (2019) uses self-supervision to improve domain generalization. Gidaris et al. (2019), the most related work to ours, also shows that self-supervision improves few-shot learning. Although this initial result is the same, we further show these benefits on smaller datasets with harder recognition problems (fine-grained classification) and with deeper models. Moreover, we also present a novel analysis of the impact of the domain of unlabeled data, and a new, simple approach to automatically select similar-domain unlabeled data for self-supervision.

## 3 METHOD

We adopt the commonly used setup for few-shot learning where one is provided with labeled training data for a set of *base* classes $\mathcal{D}_b$ and a much smaller training set (typically 1-5 examples per class) for *novel* classes $\mathcal{D}_n$. The goal of the few-shot learner is to learn representations on the base classes that lead to good generalization on novel classes. Although in theory the base classes are assumed to have a large number of labeled examples, in practice this number can be quite small for novel domains, *e.g.* less than 5000 examples for the birds dataset (Welinder et al., 2010), making it challenging to learn a generalizable representation.

Our framework, as seen in Figure 1, combines *meta-learning* approaches for few-shot learning that rely on supervised labels with *self-supervised learning* on unlabeled images. Denote a training dataset as $\{(x_i, y_i)\}_{i=1}^n$ consisting of pairs of images $x_i \in \mathcal{X}$ and labels $y_i \in \mathcal{Y}$. A feed-forward convolutional network $f(x)$ maps the input to an embedding space, which is then mapped to the label space using a classifier $g$. The overall mapping from the input to the label can be written as $g \circ f(x) : \mathcal{X} \to \mathcal{Y}$. Learning consists of estimating functions $f$ and $g$ that minimize a loss $\ell$ on the

predicted labels on the training data along with suitable regularization $\mathcal{R}$ over the functions $f$ and $g$. This can be written as:

$$\mathcal{L}_s := \sum_i \ell\big(g \circ f(x_i), y_i\big) + \mathcal{R}(f, g).$$

A commonly used loss is the cross-entropy loss and a regularizer is the $\ell_2$ norm of the parameters of the functions. For transfer $g$ is discarded and relearned on training data for novel classes.

In addition, we consider self-supervised losses $\mathcal{L}_{ss}$ based on labeled data $x \to (\hat{x}, \hat{y})$ that can be derived from inputs $x$ alone. Figure 1 shows two examples: the *jigsaw task* rearranges the input image and uses the index of the permutation as the target label, while the *rotation task* uses the angle of the rotated image as the target label. In our framework, a separate function $h$ is used to predict these labels from the shared feature backbone $f$ with a self-supervised loss:

$$\mathcal{L}_{ss} := \sum_i \ell\big(h \circ f(\hat{x}_i), \hat{y}_i\big).$$

Our final loss function combines the two losses: $\mathcal{L} := \mathcal{L}_s + \mathcal{L}_{ss}$, and thus the self-supervised losses act as a data-dependent regularizer for representation learning. Note that the domain of images used for supervised and self-supervised losses denoted by $\mathcal{D}_s$ and $\mathcal{D}_{ss}$ need not be identical. In particular we often would like to use significantly larger sets of images for self-supervised learning from related domains. Below we describe various losses we consider for image classification tasks.

## 3.1 SUPERVISED LOSSES ($\mathcal{L}_s$)

Most of our results are presented using a meta-learner based on prototypical networks (Snell et al., 2017) that perform episodic training and testing over sampled datasets in stages called meta-training and meta-testing. During meta-training, we randomly sample $N$ classes from the base set $\mathcal{D}_b$, then we select a support set $\mathcal{S}_b$ with $K$ images per class and another query set $\mathcal{Q}_b$ with $M$ images per class. We call this an $N$-way $K$-shot classification task. The embeddings are trained to predict the labels of the query set $\mathcal{Q}_b$ conditioned on the support set $\mathcal{S}_b$ using a nearest mean (prototype) model. The objective is to minimize the prediction loss on the query set. Once training is complete, given the novel dataset $\mathcal{D}_n$, class prototypes are recomputed for classification and query examples are classified based on the distances to the class prototypes.

Prototypical networks are related to distance-based learners such as matching networks (Vinyals et al., 2016) or metric-learning based on label similarity (Koch et al., 2015). In the Appendix we also present few-shot classification results using a gradient-based meta-learner called MAML (Finn et al., 2017), and one trained with a standard cross-entropy loss on the base classes. We also present standard classification results where the test set contains images from the same base categories.

## 3.2 SELF-SUPERVISED LOSSES ($\mathcal{L}_{ss}$)

We consider two losses motivated by a recent large-scale comparison of the effectiveness of self-supervised learning tasks (Goyal et al., 2019) described below:

*Jigsaw puzzle task loss.* In this case the input image $x$ is tiled into $3 \times 3$ regions and permuted randomly to obtain an input $\hat{x}$. The target label $\hat{y}$ is the index of the permutation. The index (one of 9!) is reduced to one of 35 following the procedure outlined in Noroozi & Favaro (2016), which grouped the possible permutations based on hamming distance to control the difficulty of the task.

*Rotation task loss.* We follow the method of Gidaris et al. (2018) where the input image $x$ is rotated by an angle $\theta \in \{0°, 90°, 180°, 270°\}$ to obtain $\hat{x}$ and the target label $\hat{y}$ is the index of the angle.

In both cases we use the cross-entropy loss between the target and prediction.

## 3.3 STOCHASTIC SAMPLING AND TRAINING

When the source of images used for SSL and meta-learning are identical the same batch of images are used for computing $\mathcal{L}_s$ and $\mathcal{L}_{ss}$. For experiments investigating domain shifts (Sec. 4.2), where SSL and meta-learner are trained on different distribution of images, a different batch (size of 64) is used for computing $\mathcal{L}_{ss}$. After the two forward passes, one for the supervised task and one for the

self-supervised task, the two losses are combined. While other techniques exist (Chen et al., 2018; Kendall et al., 2018; Sener & Koltun, 2018), simply combining the two losses performed well.

# 4 EXPERIMENTS

We first describe the datasets and experimental details. We then present results for few-shot learning on different datasets and SSL tasks in Section 4.1. Last, we investigate the effectiveness of SSL with unlabeled data sampled from different domains in Section 4.2.

**Datasets and benchmarks**   We experiment with datasets across diverse domains: Caltech-UCSD birds (Welinder et al., 2010), Stanford cars (Krause et al., 2013), FGVC aircrafts (Maji et al., 2013), Stanford dogs (Khosla et al., 2011), and Oxford flowers (Nilsback & Zisserman, 2006). Each dataset contains between 100 to 200 classes with a few thousands of images. We also experiment with the widely-used *mini*-ImageNet (Vinyals et al., 2016) and *tiered*-ImageNet (Ren et al., 2018) benchmarks for few-shot learning. In *mini*-ImageNet, each class has 600 images, where in *tiered*-ImageNet each class has 732 to 1300 images.

We split classes within a dataset into three disjoint sets: *base, val*, and *novel*. For each class all the images in the dataset are used in the corresponding set. A model is trained on the base set of categories, validated on the val set, and tested on the novel set of categories given a few examples per class. For birds, we use the same split as Chen et al. (2019), where {*base, val, novel*} sets have {100, 50, 50} classes respectively. The same ratio is used for the other four fine-grained datasets. We use the original splits for *mini*-ImageNet and *tiered*-ImageNet.

We also present results on a setting where the base set is "degraded" either by reducing the resolution, removing color, or the number of training examples to 20% of the original set. This allows us to study the effectiveness of SSL on even smaller datasets and as a function of the difficulty of the task. Further details of the datasets are in the Appendix A.1.

**Meta-learners and feature backbone**   We follow the best practices and use the codebase for few-shot learning described in Chen et al. (2019). In particular, we use a prototypical network (Snell et al., 2017) with a ResNet18 (He et al., 2016) as the feature backbone. Their experiments found this to often be the best performing; however, we also experiment our method with other meta-learners such as MAML (Finn et al., 2017) and softmax classifiers.

**Learning and optimization**   We use 5-way (classes) and 5-shot (examples per-class) with 16 query images for training. For experiments using 20% of labeled data, we use 5 query images for training since the minimum number of images per class is 10. The models are trained with ADAM (Kingma & Ba, 2015) with a learning rate of 0.001 for 60,000 episodes. We report the mean accuracy and 95% confidence interval over 600 test experiments. In each test episode, $N$ classes are selected from the novel set, and for each class 5 support images and 16 query images (or 5 query images when using 20% labeled data) are used. We report results for $N = \{5, 20\}$ classes.

**Image sampling and data augmentation**   Data augmentation has a significant impact on few-shot learning performance. We follow the data augmentation procedure outlined in Chen et al. (2019) which resulted in a strong baseline performance of meta-learning. For label and rotation predictions, images are first resized to 224 pixels for the shorter edge while maintaining the aspect ratio, from which a central crop of 224×224 is obtained. For jigsaw puzzles, we first randomly crop 255×255 region from the original image with random scaling between [0.5, 1.0], then split into 3×3 regions, from which a random crop of size 64×64 is picked. While it might appear that with self-supervision the model effectively sees more images, we wish to point out that all the experiments use significant data augmentation, including cropping, flipping, and color jittering, which also significantly increases the amount of training data. The gains from the self-supervised auxiliary tasks are over and above any gains from data augmentation alone. More experimental details are in Appendix A.5.

**Other experimental results**   In Appendix A.3, we show the benefits of using self-supervision for *standard* classification tasks when training the model from scratch. We further visualize these models in Appendix A.4 to show that models trained with self-supervision tend to avoid accidental correlation of background features to class labels.

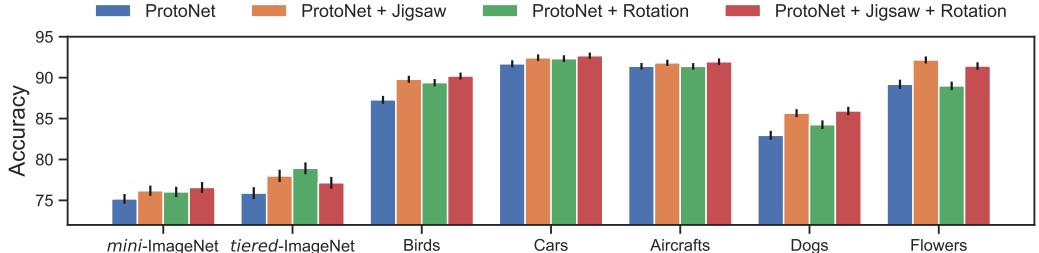

Figure 2: **Benefits of SSL for few-shot learning tasks.** We show the accuracy over the ProtoNet baseline of using different SSL tasks. The jigsaw task results in an improvement of the 5-way 5-shot classification accuracy for a ProtoNet across datasets. Combining SSL tasks can be beneficial for some datasets. Here SSL was performed on images within the base classes only. See Table 2 in the Appendix for a tabular version and results for 20-way 5-shot classification.

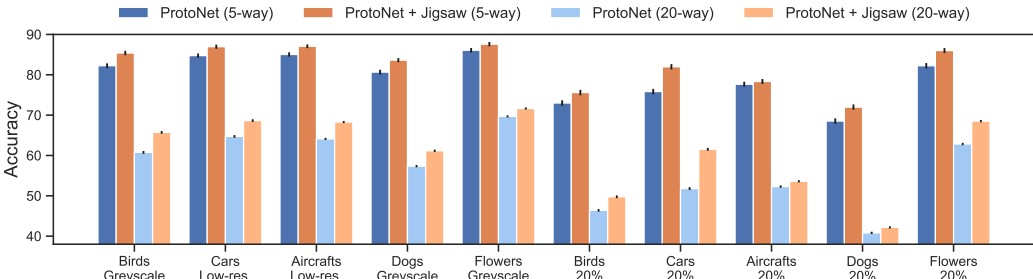

Figure 3: **Benefits of SSL for *harder* few-shot learning tasks.** We show the accuracy of using jigsaw puzzle task over ProtoNet baseline on harder versions of the datasets. We see that SSL is effective even on smaller datasets and the relative benefits are higher.

## 4.1 RESULTS ON FEW-SHOT LEARNING

**Self-supervised learning improves few-shot learning** Figure 2 shows the accuracies of various models for 5-way 5-shot classification on few-shot benchmarks. Our ProtoNet baseline matches the results of the *mini*-ImageNet and birds datasets presented in Chen et al. (2019) (in their Table A5). Our results show that jigsaw puzzle task improves the ProtoNet baseline on all seven datasets. Specifically, it reduces the *relative error rate* by 4.0%, 8.7%, 19.7%, 8.4%, 4.7%, 15.9%, and 27.8% on *mini*-ImageNet, *tiered*-ImageNet, birds, cars, aircrafts, dogs, and flowers datasets respectively. Predicting rotations also improves the ProtoNet baseline on most of the datasets, except for aircrafts and flowers. We speculate this is because most flower images are symmetrical, and airplanes are usually horizontal, making the rotation task too hard or too trivial to benefit main task. In addition, combining these two SSL tasks can be beneficial sometimes. Our results also echo those of Gidaris et al. (2019) who find that the rotation task improves on *mini*- and *tiered*-ImageNet. A tabular version of the figure and the results for 20-way classification are included in Table 2 in Appendix A.2.

**Gains are larger for harder tasks** Figure 3 shows the performance on the degraded version of the same datasets (first five groups). For cars and aircrafts we use low-resolution images where the images are down-sampled by a factor of *four* and up-sampled back to 224×224 with bilinear interpolation. For natural categories we discard color. Low-resolution images are considerably harder to classify for man-made categories while color information is most useful for natural categories (Su & Maji, 2017). On birds and dogs datasets, the improvements using self-supervision (3.2% and 2.9% on 5-way 5-shot) are higher compared to color images (2.5% and 2.7%), similarly on the cars and aircrafts datasets with low-resolution images (2.2% and 2.1% vs. 0.7% and 0.4%). We also conduct an experiment where only 20% of the images in the base categories are used for both SSL and meta-learning (last five groups in Figure 3). This results in a much smaller training set than standard few-shot benchmarks: 20% of the birds dataset amounts to only roughly 3% of the popular *mini*-ImageNet dataset. We find larger benefits from SSL in this setting. For example, the gain from the jigsaw puzzle loss for 5-way 5-shot car classification increases from 0.7% (original dataset) to 7.0% (20% training data).

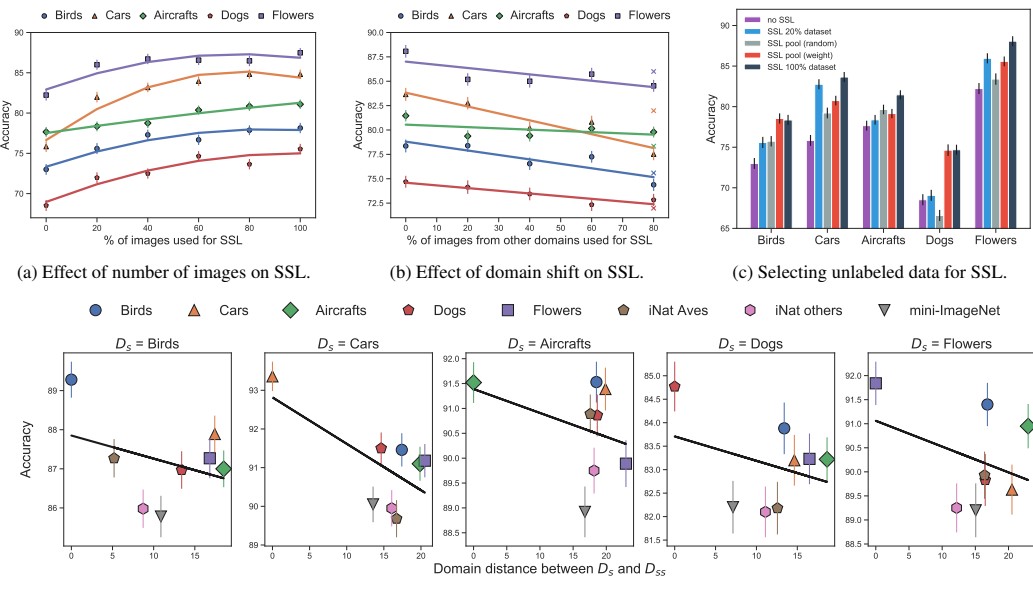

(d) Effectiveness of SSL as a function of domain distance between $\mathcal{D}_s$ and $\mathcal{D}_{ss}$ (shown on top).

Figure 4: **Effect of size and domain of SSL on 5-way 5-shot accuracy.** **(a)** More unlabeled data from the same domain for SSL improves the performance of the meta-learner. **(b)** Replacing a fraction (x-axis) of the images with those from other domains makes SSL less effective. **(c)** A comparison of approaches to sample images from a pool for SSL. With random sampling, the extra unlabeled data often hurts the performance, while those sampled using the "domain weighs" improves performance on most datasets. When the pool contains images from related domains, *e.g.*, birds and dogs, the performance using selected images matches that of using 100% within-dataset images. **(d)** Accuracy of the meta-learner with SSL on different domains as function of distance between the supervised domain $\mathcal{D}_s$ and the self-supervised domain $\mathcal{D}_{ss}$. See text for details.

**Improvements generalize to other meta-learners** We combine SSL with other meta-learners and find the combination to be effective. In particular we use MAML (Finn et al., 2017) and a standard feature extractor trained with cross-entropy loss (softmax). We observe that the average 5-way 5-shot accuracies across five fine-grained datasets for softmax, MAML, and ProtoNet improve from 85.5%, 82.6% and 88.5% to 86.6%, 83.8% and 90.4% respectively when combined with the jigsaw puzzle task. Detailed results are shown in Table 4 in Appendix A.2.

**Self-supervision alone is not enough** SSL alone significantly lags behind supervised learning in our experiments. For example, a ResNet18 trained with SSL alone achieve 32.9% (w/ jigsaw) and 33.7% (w/ rotation) 5-way 5-shot accuracy averaged across five fine-grained datasets. While this is better than a random initialization (29.5%), it is dramatically worse than one trained with a simple cross-entropy loss (85.5%) on the labels (Details in Table 2). Surprisingly, we found that initialization with SSL followed by meta-learning did not yield improvements over meta-learning starting from random initialization, supporting the view that SSL acts as a feature regularizer.

## 4.2 SELECTING DOMAINS FOR SELF-SUPERVISION

A promising direction of research in SSL is by scaling it to massive unlabeled datasets which are readily available for some domains. *However, does more unlabeled data always help for a task in hand?* This question hasn't been sufficiently addressed in the literature as most prior works study the effectiveness of SSL on a curated set of images, such as ImageNet, and their transferability to a handful of tasks. In this section we conduct a series of experiments to characterize effect of size and distribution $\mathcal{D}_{ss}$ of images used for SSL in the context of few-shot learning on a domain $\mathcal{D}_s$.

First we investigate if SSL on unlabeled data from the same domain improves the meta-learner. We use 20% of the images in the base categories for meta-learning identical to setting in Figure 3. The labels of the remaining 80% data are withheld and only the images are used for SSL. We systematically vary the number of images used by the SSL from 20% to 100%. Results are presented

in Figure 4a. The accuracy improves with the size of the unlabeled set with diminishing returns. Note that 0% corresponds to no SSL and 20% corresponds to using only the labeled images for SSL.

Figure 4b shows an experiment where a fraction of the unlabeled images are replaced with images from other datasets. For example 20% along the x-axis for birds indicate that 20% of the images in base set are replaced by images drawn uniformly at random from other datasets. Since the number of images used for SSL are identical, the x-axis from left to right represents increasing amounts of domain shifts between $\mathcal{D}_s$ and $\mathcal{D}_{ss}$. We observe that the effectiveness of SSL decreases as the fraction of out-of-domain images increases. *Importantly, training with SSL on the available 20% within domain images (shown as crosses) is often (3 out of 5 datasets) better than increasing the set of images by five times to include out of domain images.*

Figure 4d analyzes how the *magnitude* of the domain shift affects performance. Here all the images in base classes are used (instead of using 20% data in previous experiment), and all the images for SSL are sampled from different datasets. In addition to the five datasets (birds, cars, aircrafts, dogs, flowers), we include 1) 5000 images of Aves from iNaturalist 2018 dataset (iNat) (Van Horn et al., 2018), 2) 5000 images of other super-classes from iNat, and 3) 5760 images sampled from *mini-ImageNet*. These domains are roughly of the same sizes (between 5k to 10k images each) but exhibit varying degrees of domain shifts. To compute a "distance" between a pair of domains, we first estimate an embedding for each domain $\mathcal{D}$ using the mean image feature: $\Phi(\mathcal{D}) = \sum_{x_i \in \mathcal{D}} \phi(x_i)/N$, where $N$ is the number of images within $\mathcal{D}$, and use the Euclidean distance between their embeddings. Here $\phi(x)$ is an image embedding from a ResNet101 network pretrained on ImageNet. Figure 4d shows the accuracy of the meta-learner when combined with SSL on different domains as a function of the domain distance between $\mathcal{D}_s$ and $\mathcal{D}_{ss}$. Once again we see that the effectiveness of SSL decreases with the distance from the supervised domain across all datasets.

Based on the above analysis we propose a simple method to select images for SSL from a large, generic pool of unlabeled images in a dataset dependent manner. We use a "domain weighted" model to select the top images based on a domain classifier, in our case a binary logistic regression model trained with images from the source domain $\mathcal{D}_s$ as the positive class and from pool $\mathcal{D}_p$ as the negative class based on ResNet101 image features. The top images are selected according to the ratio $p(x \in \mathcal{D}_s)/p(x \in \mathcal{D}_p)$. Note that these *importance weights* account for the domain shift. We evaluate this approach using a pool of images from all five datasets as well as 30000 images each from Aves of iNat, other super-classes of iNat and *mini*-ImageNet. For each dataset we use 20% of the labeled images for meta-learning, then use the pool in a *leave-one-out* manner: *all* images from a dataset are excluded from the pool to simulate a scenario where one does not have additional within-domain images. The compositions of the selected images are shown in Appendix A.6.

Figure 4c shows results across datasets with meta-learner trained on 20% labeled examples. SSL with 20% and 100% dataset denotes baselines when SSL uses 20% and 100% of the images within domain. The latter is a reference "upper bound". SSL pool "(random)" and "(weight)" denote two approaches for selecting the remaining 80% of images from the pool. The former selects images uniformly at random, which is detrimental for cars, dogs, and flowers. For dogs the performance is even worse than no SSL. The pool selected according to the *importance weights* provides significant improvements on birds and dogs as the pool has relevant images, and outperforms the random selection on most datasets. These results are likely to improve by collecting an even larger pool.

## 5 CONCLUSION

We have shown that self-supervision improves transferability of representations on few-shot learning tasks across a range of different domains. Surprisingly, we found that self-supervision is more beneficial for more challenging problems, especially when the number of images used for self-supervision is small, orders of magnitude smaller than previously reported results. This has a practical benefit that the images within small datasets can be used for self-supervision without relying on a large-scale external dataset. We have also shown that additional unlabeled images can improve performance only if they are from the *same or similar* domains. Finally, for domains where unlabeled data is limited, we present a novel, simple approach to automatically identify such similar-domain images from a larger pool.

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

# A  APPENDIX

## A.1  DATASET DETAILS

The statistics of various datasets used in our experiments are shown in Table 1. We show results on several fine-grained classification datasets in addition to standard few-shot benchmarks such as *mini-* and *tiered*-ImageNet. Notably, the fine-grained datasets are significantly smaller.

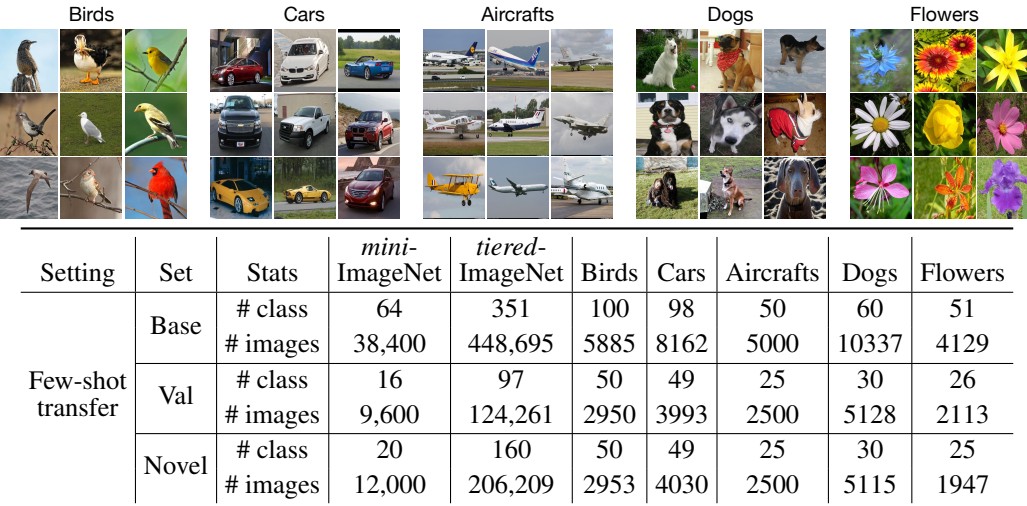

| Setting | Set | Stats | *mini*-ImageNet | *tiered*-ImageNet | Birds | Cars | Aircrafts | Dogs | Flowers |
|---|---|---|---|---|---|---|---|---|---|
| Few-shot transfer | Base | # class | 64 | 351 | 100 | 98 | 50 | 60 | 51 |
| | | # images | 38,400 | 448,695 | 5885 | 8162 | 5000 | 10337 | 4129 |
| | Val | # class | 16 | 97 | 50 | 49 | 25 | 30 | 26 |
| | | # images | 9,600 | 124,261 | 2950 | 3993 | 2500 | 5128 | 2113 |
| | Novel | # class | 20 | 160 | 50 | 49 | 25 | 30 | 25 |
| | | # images | 12,000 | 206,209 | 2953 | 4030 | 2500 | 5115 | 1947 |

Table 1: **Example images and dataset statistics**. For few-shot learning experiments the classes are split into *base*, *val*, and *novel* set. Image representations learned on *base* set are evaluated on the novel set while *val* set is used for cross-validation. These datasets vary in the number of classes but are orders of magnitude smaller than ImageNet dataset.

## A.2  RESULTS ON FEW-SHOT LEARNING

Table 2 shows the performance of ProtoNet with different self-supervision on seven datasets. We also test the accuracy of the model on novel classes when trained *only* with self-supervision on the base set of images. Compared to the randomly initialized model ("None" rows), training the network to predict rotations gives around 2% to 21% improvements on all datasets, while solving jigsaw puzzles only improves on aircrafts and flowers. However, these numbers are significantly worse than learning with supervised labels on the base set, in line with the current literature.

Table 3 shows the performance of ProtoNet with *jigsaw puzzle loss* on harder benchmarks. The results on degraded version of the datasets are shown in the top part, and the bottom part shows the results of using only 20% of the images in the base categories. The gains using SSL are higher in this setting.

Table 4 compares meta-learners based on a ResNet18 network trained with and without *jigsaw puzzle loss*. Self-supervision improves the performance across different meta-learners and different datasets. ProtoNet trained with self-supervision is the best model across all datasets.

| Loss | mini-ImageNet | tiered-ImageNet | Birds | Cars | Aircrafts | Dogs | Flowers |
|---|---|---|---|---|---|---|---|
| | | | 5-way 5-shot | | | | |
| ProtoNet | 75.2±0.6 | 75.9±0.7 | 87.3±0.5 | 91.7±0.4 | 91.4±0.4 | 83.0±0.6 | 89.2±0.6 |
| ProtoNet + Jigsaw | 76.2±0.6 | 78.0±0.7 | 89.8±0.4 | 92.4±0.4 | 91.8±0.4 | 85.7±0.5 | **92.2±0.4** |
| *Rel. error reduction* | *4.0%* | *8.7%* | *19.7%* | *8.4%* | *4.7%* | *15.9%* | *27.8%* |
| Jigsaw | 25.6±0.5 | 24.9±0.4 | 25.7±0.5 | 25.3±0.5 | 38.8±0.6 | 24.3±0.5 | 50.5±0.7 |
| ProtoNet + Rotation | 76.0±0.6 | **78.9±0.7** | 89.4±0.4 | 92.3±0.4 | 91.4±0.4 | 84.3±0.5 | 89.0±0.5 |
| Rotation | 51.4±0.7 | 50.7±0.8 | 33.1±0.6 | 29.4±0.5 | 29.5±0.5 | 27.3±0.5 | 49.4±0.7 |
| ProtoNet + Jig. + Rot. | **76.6±0.7** | 77.2±0.7 | **90.2±0.4** | **92.7±0.4** | **91.9±0.4** | **85.9±0.5** | 91.4±0.5 |
| None | 31.0±0.5 | 28.9±0.5 | 26.7±0.5 | 25.2±0.5 | 28.1±0.5 | 25.3±0.5 | 42.3±0.8 |
| | | | 20-way 5-shot | | | | |
| ProtoNet | 46.6±0.3 | 49.7±0.4 | 69.3±0.3 | 78.7±0.3 | 78.6±0.3 | 61.6±0.3 | 75.4±0.3 |
| ProtoNet + Jigsaw | 47.8±0.3 | **52.4±0.4** | 73.7±0.3 | 79.1±0.3 | **79.1±0.2** | 65.4±0.3 | **79.2±0.3** |
| Jigsaw | 9.2±0.2 | 7.5±0.1 | 8.1±0.1 | 7.1±0.1 | 15.4±0.2 | 7.1±0.1 | 25.7±0.2 |
| ProtoNet + Rotation | 48.2±0.3 | **52.4±0.4** | 72.9±0.3 | **80.0±0.3** | 78.4±0.2 | 63.4±0.3 | 73.9±0.3 |
| Rotation | 27.4±0.2 | 25.7±0.3 | 12.9±0.2 | 9.3±0.2 | 9.8±0.2 | 8.8±0.1 | 26.3±0.2 |
| ProtoNet + Jig. + Rot. | **49.0±0.3** | 51.2±0.4 | **75.0±0.3** | 79.8±0.3 | 79.0±0.2 | **66.2±0.3** | 78.6±0.3 |
| None | 10.8±0.1 | 11.0±0.2 | 9.3±0.2 | 7.5±0.1 | 8.9±0.1 | 7.8±0.1 | 22.6±0.2 |

Table 2: **Performance on few-shot transfer task.** The mean accuracy (%) and the 95% confidence interval of 600 randomly chosen test experiments are reported for various combinations of loss functions. The top part shows the accuracy on 5-way 5-shot classification tasks, while the bottom part shows the same on 20-way 5-shot. Adding self-supervised losses to the ProtoNet loss improves the performance on all seven datasets on 5-way classification results. On 20-way classification, the improvements are even larger. The last row indicates results with a randomly initialized network. The top part of this table corresponds to Figure 2.

| Loss | Birds Greyscale | Cars Low-resolution | Aircrafts Low-resolution | Dogs Greyscale | Flowers Greyscale |
|---|---|---|---|---|---|
| | | | 5-way 5-shot | | |
| ProtoNet | 82.2±0.6 | 84.8±0.5 | 85.0±0.5 | 80.7±0.6 | 86.1±0.6 |
| ProtoNet + Jigsaw | 85.4±0.6 | 87.0±0.5 | 87.1±0.5 | 83.6±0.5 | 87.6±0.5 |
| | | | 20-way 5-shot | | |
| ProtoNet | 60.8±0.4 | 64.7±0.3 | 64.1±0.3 | 57.4±0.3 | 69.7±0.3 |
| ProtoNet + Jigsaw | 65.7±0.3 | 68.6±0.3 | 68.3±0.3 | 61.2±0.3 | 71.6±0.3 |
| Loss | 20% Birds | 20% Cars | 20% Aircrafts | 20% Dogs | 20% Flowers |
| | | | 5-way 5-shot | | |
| ProtoNet | 73.0±0.7 | 75.8±0.7 | 77.7±0.6 | 68.5±0.7 | 82.2±0.7 |
| ProtoNet + Jigsaw | 75.4±0.7 | 82.8±0.6 | 78.4±0.6 | 69.1±0.7 | 86.0±0.6 |
| | | | 20-way 5-shot | | |
| ProtoNet | 46.4±0.3 | 51.8±0.4 | 52.3±0.3 | 40.8±0.3 | 62.8±0.3 |
| ProtoNet + Jigsaw | 49.8±0.3 | 61.5±0.4 | 53.6±0.3 | 42.2±0.3 | 68.5±0.3 |

Table 3: **Performance on few-shot transfer task with degraded inputs or less training data.** Accuracies are reported on novel set for 5-way 5-shot and 20-way 5-shot classification with degraded inputs, and with a subset (20%) of the images in the base set. The loss of color or resolution, and the smaller training set size make the tasks more challenging as seen by the drop in the performance of the ProtoNet baseline. However the improvements of using the *jigsaw puzzle loss* are higher in comparison to the results presented in Table 2.

| Loss | Birds | Cars | Aircrafts | Dogs | Flowers |
|---|---|---|---|---|---|
| | 5-way 5-shot | | | | |
| Softmax | 81.5±0.5 | 87.7±0.5 | 89.2±0.4 | 77.6±0.6 | 91.3±0.5 |
| Softmax + Jigsaw | 83.9±0.5 | 90.6±0.5 | 89.6±0.4 | 77.8±0.6 | 90.9±0.5 |
| MAML | 81.2±0.7 | 86.9±0.6 | 88.8±0.5 | 77.3±0.7 | 79.0±0.9 |
| MAML + Jigsaw | 81.1±0.7 | 89.0±0.5 | 89.1±0.5 | 77.3±0.7 | 82.6±0.7 |
| ProtoNet | 87.3±0.5 | 91.7±0.4 | 91.4±0.4 | 83.0±0.6 | 89.2±0.6 |
| ProtoNet + Jigsaw | **89.8±0.4** | **92.4±0.4** | **91.8±0.4** | **85.7±0.5** | **92.2±0.4** |

Table 4: **Performance on few-shot transfer task using different meta-learners.** Using jigsaw puzzle loss gives improvements across different meta-learners on most of the datasets. ProtoNet with jigsaw loss performs the best on all five datasets.

## A.3 RESULTS ON STANDARD FINE-GRAINED CLASSIFICATION

Here we present results on standard classification task. Different from few-shot transfer learning, all the classes are seen in the training set and test set contains novel images from the same classes. We use the standard training and test splits provided in the datasets. We investigate if SSL can improve training of deep networks (*e.g.* ResNet18 network) when *trained from scratch* (*i.e.* with random initialization) using images and labels in the training set only. The accuracy using various loss functions are shown in Table 5. Training with self-supervision improves performance across datasets. On birds, cars, and dogs, predicting rotation gives 4.1%, 3.1%, and 3.0% improvements, while on aircrafts and flowers, the *jigsaw puzzle loss* yields 0.9% and 3.6% improvements.

| Loss | Birds | Cars | Aircrafts | Dogs | Flowers |
|---|---|---|---|---|---|
| Softmax | 47.0 | 72.6 | 69.9 | 51.4 | 72.8 |
| Softmax + Jigsaw | 49.2 | 73.2 | **70.8** | 53.5 | **76.4** |
| Softmax + Rotation | **51.1** | **75.7** | 70.0 | **54.4** | 73.5 |

Table 5: **Performance on standard classification task.** Per-image accuracy (%) on the test set are reported. Using self-supervision improves the accuracy of a ResNet18 network trained from scratch over the baseline of supervised training with cross-entropy (softmax) loss on all five datasets.

## A.4 VISUALIZATION OF LEARNED MODELS

To understand why the representation generalizes, we visualize what pixels contribute the most to the correct classification for various models. In particular, for each image and model we compute the gradient of the logits (predictions before softmax) for the correct class with respect to the input image. The magnitude of the gradient at each pixel is a proxy for its importance and is visualized as "saliency maps". Figure 5 shows these maps for various images and models trained with and without self-supervision on the standard classification task. It appears that the self-supervised models tend to focus more on the foreground regions, as seen by the amount of bright pixels within the bounding box. One hypothesis is that self-supervised tasks force the model to rely less on background features, which might be accidentally correlated to the class labels. For fine-grained recognition, localization indeed improves performance when training from few examples (see Wertheimer & Hariharan (2019) for a contemporary evaluation of the role of localization for few-shot learning).

## A.5 EXPERIMENTAL DETAILS

**Additional optimization details on few-shot learning** During training, especially for jigsaw puzzle task, we found it to be beneficial to *not* track the running mean and variance for the batch normalization layer, and instead estimate them for each batch independently. We hypothesize that this is because the inputs contain both full-sized images and small patches, which might have different

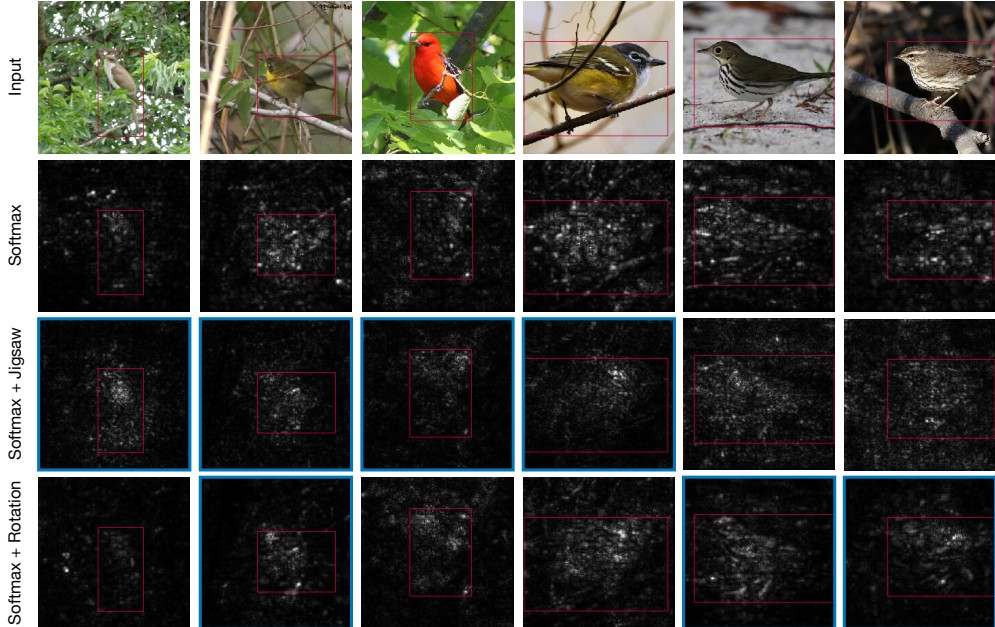

Figure 5: **Saliency maps for various images and models.** For each image we visualize the magnitude of the gradient with respect to the correct class for models trained with various loss functions. The magnitudes are scaled to the same range for easier visualization. The models trained with self-supervision often have lower energy on the background regions when there is clutter. We highlight a few examples with blue borders and the bounding-box of the object for each image is shown in red.

statistics. At test time we do the same. We found the accuracy goes up as the batch size increases but saturates at size of 64.

When training with supervised and self-supervised loss, a trade-off term $\lambda$ between the losses can be used, thus the total loss is $\mathcal{L} = (1 - \lambda)\mathcal{L}_s + \lambda\mathcal{L}_{ss}$. We find that simply use $\lambda = 0.5$ works the best, except for training on *mini-* and *tiered*-ImageNet with jigsaw loss, where we set $\lambda = 0.3$. We suspect that this is because the variation of the image size and the categories are higher, making the self-supervision harder to train with limited data. When both jigsaw and rotation losses are used, we set $\lambda = 0.5$ and the two self-supervised losses are averaged for $\mathcal{L}_{ss}$.

For training meta-learners, we use 16 query images per class for each training episode. When only 20% of labeled data are used, 5 query images per class are used. For MAML, we use 10 query images and the approximation method for backpropagation as proposed in Chen et al. (2019) to reduce the GPU memory usage. When training with self-supervised loss, it is added when computing the loss in the outer loop. We use PyTorch (Paszke et al., 2017) for our experiments.

For the domain classifier, we first obtain features from the penultimate-layer (2048 dimensional) from a ResNet101 model pretrained on ImageNet. We then train a binary logistic regression model with weight decay using LBFGS for 1000 iterations. The images from labeled dataset are the positive class and images from pool are the negative. A loss for the positive class is scaled by the inverse of its frequency to account for the significantly larger number of negative examples.

**Optimization details for standard classification**   For standard classification (Appendix A.3) we train a ResNet18 network *from scratch*. All the models are trained with ADAM optimizer with a learning rate of 0.001 for 600 epochs with a batch size of 16. We track the running statistics for the batch normalization layer for the softmax baselines following the conventional setting, *i.e.* w/o self-supervised loss, but do not track these statistics when training with self-supervision.

**Architectures for self-supervised tasks**   For jigsaw puzzle task, we follow the architecture of Noroozi & Favaro (2016) where it was first proposed. The ResNet18 results in a 512-dimensional

feature for each input, and we add a fully-connected (`fc`) layer with 512-units on top. The nine patches gives nine 512-dimensional feature vectors, which are concatenated. This is followed by a `fc` layer, projecting the feature vector from 4608 to 4096 dimensions, and a `fc` layer with 35-dimensional outputs corresponding to the 35 permutations for the jigsaw task.

For the rotation task, the 512-dimensional output of ResNet18 is passed through three `fc` layers with {512, 128, 128, 4} units. The predictions correspond to the four rotation angles. Between each `fc` layer, a `ReLU` activation and a dropout layer with dropout probability of 0.5 are added.

### A.6 Details of the dataset selected from pool

Table 6 describes the composition of the unlabeled images sampled from a generic pool for SSL experiments presented in Section 4.2 (Figure 4c). "Random sampling" uniformly samples images proportional to the size of each dataset, while the "domain weighted" one tends to pick more images from related domains. This is also indicated by the domain distance of the sampled images with respect to the source domain. The improvements are larger when the domain distance is small, *e.g.*, for birds, dogs, and flowers.

| Dataset | Labeled dataset | | | | | | | | | |
| | 20% Birds | | 20% Cars | | 20% Aircrafts | | 20% Dogs | | 20% Flowers | |
| | Selection method | | | | | | | | | |
| | random | weight | random | weight | random | weight | random | weight | random | weight |
|---|---|---|---|---|---|---|---|---|---|---|
| Birds | 1177 | 1177 | 325 | 87 | 184 | 114 | 410 | 118 | 156 | 9 |
| Cars | 303 | 2 | 1632 | 1632 | 242 | 466 | 532 | 594 | 230 | 12 |
| Aircrafts | 214 | 3 | 259 | 259 | 1000 | 1000 | 356 | 285 | 129 | 0 |
| Dogs | 413 | 19 | 554 | 657 | 334 | 251 | 2067 | 2067 | 283 | 91 |
| Flowers | 167 | 3 | 230 | 162 | 140 | 40 | 290 | 145 | 825 | 825 |
| iNat aves | 1133 | 4176 | 1599 | 792 | 957 | 815 | 1997 | 744 | 767 | 119 |
| iNat others | 1089 | 108 | 1546 | 1555 | 935 | 240 | 2048 | 964 | 793 | 2533 |
| *mini*-ImageNet | 1389 | 397 | 2017 | 3018 | 1208 | 2074 | 2637 | 5420 | 946 | 540 |
| Domain distance → | 6.69 | 0.94 | 11.42 | 9.69 | 13.41 | 11.13 | 8.72 | 3.68 | 13.38 | 4.57 |
| Accuracy (%) → | 75.7 | 78.5 | 79.2 | 80.8 | 79.7 | 79.2 | 66.6 | 74.6 | 83.4 | 85.6 |

Table 6: **Composition of the unlabeled datasets selected from a pool of images used in Figure 4c.** We list the number of images in each source datasets. We also provide domain distance between the selected images and the labeled dataset. The 5-way 5-shot accuracy of the models trained with SSL are shown in the last row. The performance correlates with domain distance: when the distribution of unlabeled data is closer to labeled data, the performance is better.

