# OpenReview forum: "When Does Self-supervision Improve Few-shot Learning?"
_ICLR.cc/2020/Conference — Reject_

### Official Review · AnonReviewer2 · 2019-10-20
**Official Blind Review #2**

**Rating:** 3

**Review:**

Summary:
There are three main contributions of the paper:
i) The authors present an empirical study of different self-supervised learning (SSL) methods in the context of self-supervised learning.
ii) They point out how SSL helps more when the dataset is harder.
iii) They point out how domain matters while using SSL for training and present a method to choose samples from an unlabeled dataset.

Strengths:
1) They confirm the results of [7] and provide additional evidence of the benefit of the self-supervised learning in the few-shot setting.  They also showcase an interesting new result that self-supervised learning helps more in case of harder problems.
2) The authors have a done a commendable job of coming up with a meaningful set of experiments by varying base-models, self-supervised methods, datasets, and few-shot learning methods in Section 4.1. This is quite a comprehensive study.
3)  The paper is well-written and well-motivated.

Weaknesses:
1) The main weakness of the paper is Section 4.2's experimental setup.

i) The definition of domain distance in not quite meaningful. Since the chosen datasets have very different classes (airplanes vs dogs etc), the average embeddings for different datasets/classes will be far from each other.  In Figure 4d, it is misleading to show a trendline that includes the same domain as that will always be 0. If that datapoint is removed the trend line is mostly flat. The authors want to present a quantifiable way to show how domain distribution affect performance on self-supervised learning methods. But this definition of domain distance is more meaningful in the domain adaptation setting (like Amazon-Office dataset used for domain adaptation (https://people.eecs.berkeley.edu/~jhoffman/domainadapt/)) as in that case the requirement is to get the embeddings close to each other for the same class but from different domains.

ii) The authors go on to create an "unlabeled pool" by combining images from many domains. Then they train a domain classifier by labeling in-domain images as positive and  labeling the images in pool as negative. Considering all images in the pool as negative is not correct as there can be images of same class in unlabeled pool.

iii) Then they choose the samples which the classifier predicts comes from the same domain. This probably succeeds as it is done on top of ResNet-101 features (that has seen all of ImageNet). This technique probably works possibly due to the ResNet being pre-trained with so many labeled classes. One baseline might have just been to choose the k-nearest neighbors (from unlabled pool)  to the average embedding of all the images of the chosen dataset. Since, it is quite easy for classifiers to detect from which dataset an image came from [6] it might be easy to choose an image from similar domain by using nearest neighbor in the embedding space.

I am not sure how well their heuristic will work for an unlabeled pool that the classifier has never seen. Additionally, a portion of the dataset has been created by combining existing datasets. Since statistics of different datasets vary a lot artificially, the creation of an unlabeled pool by combining different datasets might work in the favor of the proposed heuristic of looking at "domain distance" to choose the samples.

2) Effect of domain shift in SSL (Figure 4b) is studying an extreme case where the domain shift results in almost no common classes in the SSL training phase. It would make more sense to include datasets where at least some of the classes are shared so that self-supervised learning methods get to see some relevant classes. In practice, a self-supervised learning method would be applied on a large unlabeled pool of images. Hopefully with increasing diversity and number of images, there might be some images on which ding SSL helps the downstream few-shot task. Hence comparison with such a dataset like ImageNet/iNatrualist is important here.

Decision:
The paper presents an well thought-out empirical study on self-supervised learning for few-shot learning. But there are  major concerns with the empirical setup and methods presented in the section where they propose a method to choose samples for SSL from an unlabeled pool of images.


Minor Comments:
1)  “In contrast, we humans can quickly learn new concepts from limited training data” - Remove we.
2) “Despite recent advances, these techniques have only been applied to a few domains (e.g., entry-level classes on internet imagery), and under the assumption that large amounts of unlabeled images are available.” This is not true. Self-supervised learning methods have been used for continuous control in reinforcement learning[1, 2], cross-modal learning[3], navigation [5], action recognition[4] etc. Later on in related work the authors list many papers that use self-supervised learning in different contexts. This line should be modified to reflect how common self-supervised learning methods are in other fields as well and with less data.
3) “making the rotation task too hard or too trivial to benefit main task” - add respectively.
4) "With random sampling, the extra unlabeled data often hurts the performance, while those sampled using the “domain weighs” improves performance on most datasets." replace weighs with weights
5) [8] points out how self-supervised learning on a large dataset but with a domain shift (YFCC100M) is not as effective for pre-training as it is doing self-supervised learning on the downstream task's dataset (ImageNet). While it is a different setting it is inline with one of the main conclusions of the paper.

References:
[1] “PVEs: Position-Velocity Encoders for Unsupervised Learning of Structured State Representations” Rico Jonschkowski, Roland Hafner, Jonathan Scholz, and Martin Riedmiller.
[2] “Learning Actionable Representations from Visual Observations” Debidatta Dwibedi, Jonathan Tompson, Corey Lynch, Pierre Sermanet
[3] “Look, Listen and Learn” Relja Arandjelović, Andrew Zisserman
[4] “Self-supervised Spatiotemporal Learning via Video Clip Order Prediction” Dejing Xu, Jun Xiao, Zhou Zhao, Jian Shao, Di Xie, Yueting Zhuang.
[5] “Scaling and Benchmarking Self-Supervised Visual Representation Learning” Priya Goyal, Dhruv Mahajan, Abhinav Gupta, Ishan Misra
[6] "Unbiased Look at Dataset Bias" Antonio Torralba and Alexei A. Efros.
[7] "Boosting ´few-shot visual learning with self-supervision." Spyros Gidaris, Andrei Bursuc, Nikos Komodakis, Patrick Perez, and Matthieu Cord.
[8] "Deep clustering for unsupervised learning of visual features" Mathilde Caron, Piotr Bojanowski, Armand Joulin, and Matthijs Douze


**Experience Assessment:**

I have published one or two papers in this area.

**Review Assessment: Checking Correctness Of Derivations And Theory:**

N/A

**Review Assessment: Checking Correctness Of Experiments:**

I carefully checked the experiments.

**Review Assessment: Thoroughness In Paper Reading:**

I read the paper thoroughly.

---

> ### Author Response · Authors · 2019-11-08
> **Addressing to the concerns from R2**
>
>
> We thank R2 for noting the strengths of the paper. We wish to clarify some of the issues related to the experiments.
>
> ** Concern 1-1 the definition of domain distance is not quite meaningful **
> The trend lines in 4d are indeed not meaningful. We will update this figure. The main issues are the domains are indeed very different and measuring domain distance across different tasks and domains is an open research problem. Figure 4b, where we systematically replace part of the domain from the other datasets allows us to continuously vary the domain distance and the trend lines are more clear.
>
> ** Concern 1-2 considering all images in the pool as negative is not correct as there can be images of the same class in unlabeled pool **
> We create the pool in a leave-one-out manner, i.e. images from the target dataset are excluded. Note that the goal of the domain classifier is to simply find images (or compute their importance weights) from the pool that are similar to the source domain, and approaches based on image similarity are also likely to work.
>
> ** Concern 1-3 about using ResNet-101 features for training the domain weighted model **
> We are not sure what the reviewer is concerned about and would appreciate a clarification.
>
> Your intuition is right, in that the goal of the domain classifier is to simply predict which images are likely to come from a dataset. This might be possible to predict even with hand-crafted image features. We note that although we use a ResNet model trained on ImageNet to estimate the domain distance, most of the datasets used in our experiments do not contain images from ImageNet. Even the classes are quite disjoint from ImageNet classes.
>
> ** Concern 2 include datasets where at least some of the classes are shared for Figure 4b **
> Figure 4c conducts this experiment exactly where we create a much larger pool of image by including the iNaturalist and ImageNet datasets. Selecting the right images from this larger pool benefits the Birds and Dogs since the pool has images closer to this domain. Also note that figure 4b simulates a scenario where the pool contains a mixture of within and out of domain images of different proportions.

---

### Official Review · AnonReviewer3 · 2019-10-20
**Official Blind Review #3**

**Rating:** 3

**Review:**

Summary & Pros
- This paper proposes a few-shot learning method that uses self-supervision as an auxiliary label and trains primary and auxiliary labels via multi-task learning.
- This paper provides extensive experiments for analyzing the effect of self-supervision on various few-shot learning settings: (1) self-supervision can improve various few-shot learning algorithms, ProtoNet & MAML; (2) self-supervision with similar samples can provide more improvements.

Concerns #1: Novelty of the proposed method
- This paper uses a multi-task learning approach with self-supervision. But this approach is already used in various tasks, e.g., domain adaptation, semi-supervised learning, training GANs. Thus, the proposed method (in Section 3) using a multi-task learning objective with self-supervised losses seems to be incremental.

Concerns #2: Somewhat unsurprising experimental results
- This paper shows various experimental results, but some experiments seem to be trivial. For example, the performance gap is typically increased when learning harder tasks, e.g., when the number of training samples is decreasing, the performance gap between methods is typically increasing in a fully-supervised setting. Thus I think results in the paragraph "Gains are larger for harder tasks" might be predictable. Other examples are Figure 4a and 4b in Section 4.2 because one can easily expect that using training more in-domain samples can provide more performance gain.
- In the case of Figure 4d in Section 4.2, the authors claimed that the effectiveness of SSL decreases as the distance from the supervised domain increases. However, I think Figure 4d is not matched to the claim. For example, in the case of Dogs, better performance is achieved when using a more dissimilar domain for self-supervision except for D_s=D_ss. So I wonder how to draw the lines in Figure 4d.

Some experimental results provide meaningful messages, e.g., single self-supervision can improve performance significantly while joint self-supervision does marginally. However, the contribution of the methodology is limited and some experimental results seem to incremental.


**Experience Assessment:**

I have read many papers in this area.

**Review Assessment: Checking Correctness Of Derivations And Theory:**

I assessed the sensibility of the derivations and theory.

**Review Assessment: Checking Correctness Of Experiments:**

I assessed the sensibility of the experiments.

**Review Assessment: Thoroughness In Paper Reading:**

I read the paper at least twice and used my best judgement in assessing the paper.

---

> ### Author Response · Authors · 2019-11-08
> **Addressing to the concerns from R3**
>
>
> ** Concern 1 about novelty **
> Our novelty lies in the result that self-supervised learning improves few-shot learning on small datasets, with state-of-the-art deep networks, and on fine-grained domains. We also systematically estimate the effect of domain shifts through extensive experiments across datasets. These results are likely of interest to the community.
>
> ** Concern 2 about unsurprising results **
> We do think that some of these results are surprising. For example:
> - The harder tasks we consider use only 20% of the training data, amounting to only one or two thousand images. It is surprising that SSL is effective with such small-sized datasets since the conventional wisdom is that SSL requires orders of magnitude more data to be effective.
> - Despite numerous advances, SSL still dramatically lags behind supervised learning. Ours is one of the first results showing that SSL provides improvements on top of supervised learning in the few-shot setting.
> - The results in Figure 4a and 4b are perhaps less surprising, but we are not aware of any work that systematically conducted these experiments. For example, more data but from a different domain makes SSL less effective, even worse than no SSL. Thus simply relying on the few within-domain images is the best strategy for novel domains. This result is of practical significance for learning in domains where even unlabeled data is hard to obtain.
>
> Figure 4d is indeed confusing. The difficulty stems from how to measure the domain distance, which is an open research problem. The main result still holds, i.e., domain shift makes the SSL less effective (as also seen in Figure 4b), but the right way to measure a domain distance that is predictive of transfer is an open question. We will remove the trend lines in the figures. Also see the response to R2.

---

### Official Review · AnonReviewer1 · 2019-10-25
**Official Blind Review #1**

**Rating:** 8

**Review:**

Summary: This paper considers the addition of self-supervised learning techniques in the few-shot learning setting. Extensive experiments are done to show that it can be helpful, including in cases where the labeled data is corrupted. The paper also considers the domain mismatch issue where unlabeled images come from a different domain.

Review: This paper is thorough and clearly written. Applying self-supervised learning techniques to the few-shot learning regime is a simple idea, and this paper clearly shows that it can be beneficial. It includes extensive experiments on a wide variety of image datasets, including many additional studies in the appendix. The only possible criticisms of this paper are that it is limited to the image domain (as are most few-shot learning/self-supervised learning studies, so we can probably ignore this) and that it does not produce a huge delta in understanding compared to Gidaris et al. (2019). Gidaris et al. (2019) was posted to arxiv in June, I'm not sure if this work was also on arxiv around the same time, and even if it wasn't I'm not sure what to consider "concurrent". However, as the authors note they include additional experimental settings (like the domain-selection idea) that are not in Gidaris et al. (2019), so the works are somewhat complementary. The only other comment I have is that the paper is quite large in scope for a conference submission and as a result there are many details and experiments that are left for the appendix. I could also see the domain selection experiments constituting their own submission. For example the definition of the "'distance' between a pair of domains" is only introduced in passing in the midst of Section 4.2 covering domain shift experiments, and the method for training a domain classifier is similarly only mentioned in passing in 4.2.  Of course, it is not really valid to criticize a paper for being too exhaustive. Overall, I recommend acceptance.

Specific comments:
- Truly a minor suggestion but I suggest moving Figure 1 to the top of page 2.
- Snell et al. (2017) needs a \citep
- You ought to cite "S4L: Self-Supervised Semi-Supervised Learning", which is related to your discussion of connections between self- and semi-supervised learning (though not few-shot).
- The paragraph beginning "The focus of most prior work..." in the Related Work section provides a nice framing of your work and so might make more sense in the introduction.
- "we consider self-supervised losses based on labeled data ... that can be derived from inputs x alone" All labels (can be derived from in the inputs x alone, given an oracle (or human labeler). I think you mean "that can be derived automatically without any human labeling".
- You state "Our final loss function combines the two losses". Is there no scalar multiplier on either loss term to trade-off the importance of each?
- The gains from the self-supervised auxiliary tasks are over and above any gains from data augmentation alone. More experimental details are in Appendix A.5." I don't see any information to substantiate the claim that the self-supervised tasks result in a bigger improvement than data augmentation alone, can you provide those details?
- "However, does more unlabeled data always help for a task in hand?" This question actually was addressed somewhat in "Realistic Evaluation of Semi-Supervised Learning Algorithms", see sections 4.4 and 4.5 therein.

**Experience Assessment:**

I have published in this field for several years.

**Review Assessment: Checking Correctness Of Derivations And Theory:**

N/A

**Review Assessment: Checking Correctness Of Experiments:**

I assessed the sensibility of the experiments.

**Review Assessment: Thoroughness In Paper Reading:**

N/A

---

> ### Author Response · Authors · 2019-11-08
> **Thank you for your positive comments**
>
>
> ** Comparing to Gidaris et al. **
> Gidaris et al. is a concurrent work of ours. While there are many differences (e.g. they use 64*64 images and a WRN-28 model while we use 224*224 images and a ResNet18 model), the improvements on mini-ImageNet and tiered-ImageNet on 5-way 5-shot classification are in the same range as ours. We also additionally report results on several fine-grained domains.
>
> ** Scalar multiplier **
> We found using equal weights for the two losses works well (same conclusion as Gidaris et al.), so we did not include the scalar term in the main text. The only exception is on mini-ImageNet and tiered-ImageNet which is explained in Appendix A.5.
>
> ** “The gains from the self-supervised auxiliary tasks are over and above any gains from data augmentation alone” **
> We meant that even after extensive data augmentation (cropping, jittering, flipping, etc.), SSL provides consistent improvements. A.5 describes the details of the data augmentation. Apologies for the confusion.
>
> As an additional experiment we ran the baselines without data augmentation as shown below. The results are worse, but SSL provides consistent improvements.
> ═════════════════════════════════════════════
>                                                                             Dataset
> Method                                    Birds     Cars    Aircrafts   Dogs    Flowers
> ────────────────────────────────
> ProtoNet (w/o aug)                 74.0      75.7       78.1       59.3        80.7
> ProtoNet+Jigsaw (w/o aug)   77.7      83.5       80.7       60.3        86.8
> ProtoNet (w/ aug)                    87.3      91.7       91.4       83.0        89.2
> ProtoNet+Jigsaw (w/ aug)      89.8      92.4       91.8       85.7        92.2
> ═════════════════════════════════════════════
>
> ** Related work **
> Indeed the two works are related and complementary to ours;  Zhai et al. show that self-supervised and semi-supervised learning methods can be complementary when a large unlabeled dataset (ImageNet) is used. Oliver et al. investigate the effect of extra unlabeled data and the domain mismatch for semi-supervised learning. However, it is not clear if the same conclusion holds for few-shot learning, the main focus of this paper. We also focus our evaluation on several fine-grained domains. We will include a discussion in the related work.

---

> > ### Comment · AnonReviewer1 · 2019-11-15
> > **Response**
> >
> > Thanks for your comments and updates.

---

### Decision · Program_Chairs · 2019-12-19

**Decision:**

Reject

**Comment:**

Reviewers agree that this paper contains interesting results and simple, but good ideas. However, a few severe concerns were raised by reviewers. Most prominent one was the experiment set up - authors use a pre trained ResNet101 (which has seen many classes of Imagenet) for testing which makes is unclear how well their proposed method would work for unlabeled pool of dataset that classifiers has never seen.
 While authors claim that their the dataset used for testing was disjoint from Imagenet, a reviewer pointed out that dogs dataset, bird datasets both state that they overlap with Imagenet. A few other concerns are raised (need more meaningful metric in Figure 4d, which wasn’t addressed in rebuttal). We look forward to seeing an improved version of the paper in your future submissions.